# Pharmacokinetic Profiling Using ^3^H-Labeled Eggshell Membrane and Effects of Eggshell Membrane and Lysozyme Oral Supplementation on DSS-Induced Colitis and Human Gut Microbiota

**DOI:** 10.3390/ijms26189102

**Published:** 2025-09-18

**Authors:** Miho Shimizu, Wataru Sugai, Eri Ohto-Fujita, Aya Atomi, Norio Nogawa, Koichi Takamiya, Hisao Yoshinaga, Yoshihide Asano, Takashi Yamashita, Shinichi Sato, Atsushi Enomoto, Nozomi Hatakeyama, Shunsuke Yasuda, Kazuya Tanaka, Tomoaki Atomi, Kenji Harada, Yukio Hasebe, Toshiyuki Watanabe, Yoriko Atomi

**Affiliations:** 1Material Health Science, Graduate School of Engineering, Tokyo University of Agriculture and Technology, Tokyo 184-8588, Japan; wata.rex999@gmail.com (W.S.); fujita.eri.rj@teikyo-u.ac.jp (E.O.-F.); s216883w@st.go.tuat.ac.jp (A.A.); yasuko927856@gmail.com (S.Y.); 2Division of Applied Chemistry, Graduate School of Engineering, Tokyo University of Agriculture and Technology, Tokyo 184-8588, Japan; toshi@cc.tuat.ac.jp; 3Isotope Science Center, The University of Tokyo, Tokyo 113-0032, Japan; nogawa@ric.u-tokyo.ac.jp; 4Institute for Integrated Radiation and Nuclear Science, Kyoto University, Osaka 590-0494, Japan; takamiya.koichi.2u@kyoto-u.ac.jp (K.T.); yoshinaga.hisao.4n@kyoto-u.ac.jp (H.Y.); 5Department of Dermatology, Tohoku University Graduate School of Medicine, Sendai 980-8574, Japan; yasano@derma.med.tohoku.ac.jp; 6Department of Dermatology, Graduate School of Medicine, The University of Tokyo, Tokyo 113-8655, Japan; yamashitat-der@h.u-tokyo.ac.jp (T.Y.); satos-der@h.u-tokyo.ac.jp (S.S.); 7Radiation Biology, Center for Disease Biology and Integrative Medicine, Graduate School of Medicine, The University of Tokyo, Tokyo 113-0033, Japan; aenomoto@m.u-tokyo.ac.jp; 8Department of Physical Therapy, Faculty of Medical Science, Teikyo University of Science, Yamanashi 409-0193, Japan; k-tanaka@ntu.ac.jp; 9Physical Therapy Major, Department of Rehabilitation, Faculty of Health Sciences, Kyorin University, Tokyo 181-8612, Japan; tatomi@ks.kyorin-u.ac.jp; 10Health Service Center, Tokyo University of Agriculture and Technology, Tokyo 183-8538, Japan; kharada@cc.tuat.ac.jp; 11Almado Inc., Tokyo 103-0022, Japan

**Keywords:** eggshell membrane (ESM), lysozyme (LYZ), dextran sulfate sodium (DSS) colitis, intestinal fibrosis, gut microbiota, *Bifidobacterium*, *Lactobacillales*, inflammatory bowel disease (IBD)

## Abstract

Eggshell membrane (ESM) is composed of approximately 90% protein. Our previous studies in healthy adults demonstrated that two months of daily ESM intake improved respiratory function, zigzag walking speed, and skin elasticity. The present study aims to address the knowledge gap regarding the in vivo effects of ESM in the context of inflammatory bowel disease (IBD). Proteomic analysis was performed on powdered ESM used as a dietary supplement. To investigate its pharmacokinetics in mice, tritium (^3^H)-labeled ESM was prepared using the ^6^Li(n,α)^3^H nuclear reaction. The therapeutic potential of ESM was further examined in a 2.0% dextran sulfate sodium (DSS)-induced murine model of IBD. In addition, fecal samples from both mice and healthy human subjects were analyzed using a modified terminal restriction fragment length polymorphism (T-RFLP) method. Lysozyme C (LYZ) was the most abundant protein (47%), followed by lysyl oxidase (12%) in ESM used in this study. ^3^H-ESM was mixed with MediGel, and orally administered to mice. Radioactivity levels were measured in blood, organs (duodenum, small intestine, large intestine, liver, kidney, lung, skin), and rectal feces at 0.5, 2, 5, 24, 48, and 72 h post-administration. Radioactivity in feces indicated excretion of undigested components, while systemic distribution suggested potential whole-body effects of ESM. Oral ESM and LYZ significantly alleviated body weight loss, diarrhea, and hematochezia in a DSS-induced murine model of IBD, leading to a significantly lower disease activity index on day 3 and showing a similar trend on day 5. Gut microbiota analysis showed increased *Bacteroidales* in the DSS group, while the ESM + DSS group maintained levels similar to the control. In humans, a double-blind, randomized controlled trial was conducted to evaluate the effects of ESM on gut microbiota in healthy adults. Participants received either ESM or placebo for 8 weeks. revealed a significant increase in alpha diversity at weeks 1 and 8 in the ESM group (*p* < 0.05), with between-group differences evident from week 1 (*p* < 0.01). ESM intake reduced *Bacteroides* and significantly increased *Bifidobacterium* and *Lactobacillales* at weeks 4 and 8. These findings suggest ESM supplementation promotes beneficial modulation of gut microbiota. These findings suggest that ESM, through its major protein components such as LYZ, may serve as a promising dietary intervention for maintaining intestinal health and mitigating inflammation in the context of IBD.

## 1. Introduction

Intestinal fibrosis is a pathological condition characterized by the excessive accumulation of extracellular matrix (ECM) due to chronic inflammation or tissue injury, leading to the hardening of the intestinal wall [1,2]. It is commonly observed in inflammatory bowel diseases (IBD) [3], ischemic colitis [4], and post-radiation therapy [5]. In Crohn’s disease and ulcerative colitis, persistent inflammation activates fibroblasts via cytokines such as transforming growth factor-beta (TGF-β), thereby promoting collagen production and accelerating fibrosis [6,7]. This pathological progression results in intestinal strictures and dysmotility, which severely impair patients’ quality of life (QOL) [8]. Since colonic fibrosis involves irreversible structural changes, early control of inflammation and the development of new therapeutic strategies are urgently needed [9].

The intestinal epithelium contains glandular structures known as crypts, which secrete proteolytic and glycolytic enzymes. Because the epithelium is constantly exposed to physical and chemical stress from food intake, epithelial cells are rapidly renewed within a few days [10]. When this repair process is insufficient, the intestinal barrier is compromised, leading to bleeding, bacterial translocation, immune cell infiltration, and the potential development of chronic inflammation and colorectal cancer [11,12]. Therefore, the intestinal epithelial basement membrane and its underlying ECM must possess both structural resilience to prevent damage and regenerative capacity to promptly repair it.

Although fibrosis was traditionally regarded as irreversible, recent insights indicate that the process can be modulated by altering the mechanical and biochemical properties of the extracellular matrix [13,14]. Eggshell membrane (ESM), a natural biomaterial historically used in Asian countries as a traditional remedy, was documented for its wound-healing properties in the ancient Chinese pharmacopeia Compendium of Materia Medica over 400 years ago. In modern times, ESM supplementation has been reported to alleviate knee and joint pain [15,16,17]. Our previous studies in healthy adults have also shown that two months of daily ESM intake improves respiratory function, zigzag walking speed, and skin elasticity [18]. Eggshell membrane and its major components, LYZ and ovotransferrin, enhance the secretion of decorin—a key endogenous antifibrotic mediator—from lung fibroblasts, and ameliorate bleomycin-induced pulmonary fibrosis. Furthermore, a 22-week supplementation study in healthy subjects demonstrated enhanced lung capacity, suggesting its translational potential from bench to clinic [19]. The antifibrotic properties of eggshell membrane components, particularly via decorin-mediated inhibition of TGF-β signaling and YAP/TAZ pathway modulation, have been discussed in recent reviews on pulmonary fibrosis [20].

Building upon these findings, the present study investigates whether daily intake of ESM supplements can alleviate ulcerative colitis. ESM consists of approximately 90% protein by dry weight, including collagens (types I, V, X), lysozyme, ovotransferrin, and other structural and functional proteins [21,22]. We conducted proteomic analysis of the ESM powder used in this study to characterize its bioactive components. Using our state-of-the-art labeling technology, which enables tagging of water-insoluble complex protein materials [23], we performed pharmacokinetic profiling with tritium (^3^H)-labeled ESM in mouse. We then examined the in vivo behavior of ESM-derived bioactive components, as well as the therapeutic effects of ESM in a dextran sulfate sodium (DSS)-induced colitis mouse model [24,25], with a focus on gut microbiota modulation as a potential mechanism.

Furthermore, a double-blind, placebo-controlled intervention trial in healthy adults demonstrated that morning and evening intake of ESM supplements improved gut microbiota balance and increased the abundance of beneficial bacteria, including *Bifidobacterium* and *Lactobacillales*. This study will provide a knowledge gap regarding the in vivo effects of ESM in humans in the context of IBD.

## 2. Results

### 2.1. Proteomic Analysis of Eggshell Membrane Powder

ESM is known to be composed of approximately 90% protein. In this study, we performed a proteomic analysis of the finely powdered eggshell membrane used in our experiments. After trypsin digestion, peptides were subjected to LC-MS/MS, and the resulting spectra were identified using the Mascot search engine in combination with the UniProt database. Figure 1 presents the major proteins identified, each accounting for more than 1% of the total protein content. Among them, LYZ and a lysyl oxidase homolog were identified with high confidence scores. In addition to structural proteins such as type X collagen, several bioactive components were also detected, including lysozyme and various protease inhibitors with potential antimicrobial functions. These results suggest that eggshell membrane functions not only as a structural support matrix but also as a multifunctional natural biomaterial with antibacterial and anti-inflammatory properties.

### 2.2. In Vivo Distribution of Tritium-Labeled Eggshell Membrane Following Oral Administration in Mice

To investigate the pharmacokinetics of eggshell membrane, tritium (^3^H)-labeled eggshell membrane powder was mixed with dosing jelly and orally administered to mice, following a previously reported method [23]. For clarity, a concise table summarizing the ARRIVE checklist has been included as Appendix A. Radioactivity distribution was measured in blood, organs, and rectal feces. Mice were dissected at 0.5, 2, 5, 24, 48, and 72 h post-administration, and radioactivity was quantified in each tissue. The results showed that absorbed components reached peak blood concentrations at 5 h post-administration (Figure 2A) and were widely distributed in the duodenum (Figure 2B), small intestine (Figure 2C), large intestine (Figure 2D), liver (Figure 2E), kidney (Figure 2F), lung (Figure 2G), and skin (Figure 2H), and rectal feces (Figure 2I). The detection of eggshell membrane-derived radioactivity in rectal feces indicates the excretion of undigested components, while the distribution of absorbed components to various organs via the bloodstream suggests that eggshell membrane, when taken as a supplement, may exert systemic effects. Lysozyme, a major component of the eggshell membrane used in this study, is known to be taken up by intestinal epithelial cells via the endocytic receptor megalin. Accordingly, the earlier absorption peaks observed in the duodenum (Figure 2B) and small intestine (Figure 2C) at 2 h post-administration, compared to the peak blood concentration at 5 h (Figure 2A), are likely attributable to megalin-mediated uptake [26]. The early peak in the lungs is thought to be due to substances that avoided the first-pass effect through oral absorption [27] and were transported to the lungs via the heart as venous blood (Figure 2G). Interestingly, the kidney showed an early peak at 2 h, preceding the serum peak at 5 h, and was likely reabsorbed and gradually excreted thereafter. This observation is intriguing because inhibition or blockade of megalin-mediated endocytosis of nephrotoxic substances has been reported as a potential therapeutic strategy against drug-induced nephrotoxicity and metabolic kidney diseases.

### 2.3. DSS-Induced Colitis Mouse Model and Eggshell Membrane Supplementation: Effects on Intestinal Pathology and Gut Microbiota

IBD, encompassing ulcerative colitis (UC) and Crohn’s disease (CD), is characterized by persistent or recurrent diarrhea, abdominal pain, and hematochezia. The dextran sulfate sodium (DSS)-induced colitis model is widely used as an animal model for IBD [24,25]. Animals treated with DSS exhibit soft stools, diarrhea, and occasionally rectal bleeding. In this study, we evaluated the effects of finely pulverized eggshell membrane powder, administered orally once daily in animal jelly to mimic human supplement intake, on intestinal pathology and gut microbiota in a DSS-induced colitis mouse model. For clarity, a concise table summarizing the ARRIVE checklist has been included as Appendix A. In nutritional studies, incorporating test substances into animal feed often complicates direct comparisons with human supplement intake in terms of dosage and timing. To address this, we adhered to a supplementation-like administration method, previously validated in a bleomycin-induced pulmonary fibrosis model [19], and calculated the dosage based on human ESM intake adjusted for body weight. Additionally, we examined LYZ—identified by our proteomic analysis as a predominant ESM protein—to clarify the cellular mechanisms underlying ESM activity. Our earlier work demonstrated that LYZ is a key component that enhances decorin secretion and mediates anti-fibrotic responses in TGF-β-stimulated lung fibroblasts [19]. The anti-inflammatory effects of ESM and LYZ intake were evaluated using the disease activity index (DAI), which is calculated as the sum of scores for body weight change, stool consistency, and fecal bleeding [28,29]. At day 3, DAI was evident and progressively increased in the DSS-induced colitis model administered with 2.0% DSS (Figure 3A). However, the increase in DAI was attenuated in mice pretreated with either ESM or LYZ compared with DSS alone (Figure 3A). Group comparisons indicated that at day 3, both ESM- and LYZ-treated groups exhibited significantly lower DAI scores (*p* < 0.01), and these lower values were maintained at days 4 and 5 (Figure 3A).

Administration of DSS in mice has been shown to induce colon shortening and intestinal damage [30]. Although the mechanism by which DSS induces colitis remains unclear, its highly negatively charged sulfate groups are thought to damage the colonic epithelium, causing erosion, barrier dysfunction, and increased permeability [31]. In this experiment, the colon (including the cecum) was excised from mice, and the linear length of the rectum and colon was measured. The results showed no statistically significant changes compared to the control group on day 3, but on day 5, all DSS-treated groups exhibited a significant shortening of colon length (n = 4, *p* < 0.01), as shown in Figure 4A with representative photos from each group. However, in the groups that consumed ESM and LYZ, although no statistically significant differences were observed in the macroscopic evaluation, a slight trend toward recovery in colon length was observed, prompting histological evaluation (Figure 4A).

Cross sections of the excised colon were prepared and histopathological evaluation of the colon of DSS-induced colitis model mice was performed by HE staining (Figure 4B). The histological index (HI), graded by mucosal damage, was significantly higher in the DSS model, whereas pretreatment with ESM or LYZ reduced the score. To further analyze collagen deposition, Sirius Red (SR) staining was performed. DSS-treated colons exhibited relatively dense signals in the submucosal area; therefore, the ratio of Sirius Red–positive staining in the submucosa relative to the mucosa was compared (Figure 4C). Oral administration of ESM or LYZ resulted in a slight reduction in the submucosa/mucosa ratio, suggesting ECM remodeling.

Given that the pathogenesis of IBD is driven by pathogenic immune responses to gut microbiota, we analyzed the intestinal microbiota using the T-RFLP method to evaluate the mitigating effects of ESM and its major component LYZ in a DSS-induced colitis model. Based on a mouse gut microbiota database, the estimated relative peak areas of operational taxonomic units (OTUs) were calculated and visualized as pie charts (Figure 5A–D). From these peak area ratios, the Shannon-Wiener diversity index was calculated [32]. The DSS group showed a tendency toward reduced microbial diversity (*p* = 0.057), whereas ESM and LYZ intake appeared to prevent this reduction (Figure 5E). The relative abundance of *Bacteroidales* (phylum *Bacteroidetes*), a dominant bacterial group, increased in the DSS group (*p* = 0.057), while the DSS + ESM group showed a trend toward normalization (*p* = 0.054), indicating a strong tendency. The group-wise variation in *Lactobacillus* (phylum *Firmicutes*) showed a trend similar to that of the Shannon-Wiener index. Recent studies have reported that soft type III collagen is more abundant in young [33] and highly regenerative tissues [34,35,36]. In our previous studies, we also observed that topical application increased type III collagen in the dermal papilla of mouse skin, contributing to improved skin elasticity [37]. A two-week intake of ESM or LYZ in mice significantly increased the Col3a1/Col1a1 gene expression ratio in the rectum (Figure 5H).

### 2.4. Randomized Controlled Trial Assessing Gut Microbiota Response to Eggshell Membrane

To investigate the effects of an ESM-based supplement on the human gut microbiota, a double-blind, randomized controlled trial was conducted in healthy individuals. Participants were assigned to either a control group receiving only excipients (without ESM) or a test group receiving the eggshell membrane supplement. Fecal samples were collected at five time points: baseline (0 weeks), and at 1, 2, 4, and 8 weeks after the start of supplementation. Gut microbiota composition was analyzed using the Nagashima method, a modified terminal restriction fragment length polymorphism (T-RFLP) technique for detecting 16S rRNA gene polymorphisms [38]. Figure 6 shows the mean relative abundances over time for 10 operational taxonomic units (OTUs): *Bifidobacterium*, *Lactobacillales*, *Bacteroides*, *Prevotella*, *Clostridium cluster IV*, *subcluster XIVa*, *cluster IX*, *cluster XI*, *cluster XVIII*, and *others*. Alpha diversity, assessed by the Shannon-Wiener diversity index, showed no significant changes over time in the control group (Figure 7A). In contrast, the eggshell membrane group exhibited a significant increase in diversity at both 1 week and 8 weeks compared to baseline (Figure 7B, *p* < 0.05). Between-group comparisons revealed a significant increase in alpha diversity in the eggshell membrane group beginning at 1 week, with highly significant differences observed at 8 weeks (** *p* < 0.01, *** *p* < 0.001; Figure 7C). This increase in diversity was associated with a relative reduction in *Bacteroides*, the most dominant genus, and a corresponding increase in other taxa (Figure 6). A detailed analysis of the 10 classified OTUs revealed that the proportions of beneficial bacteria, specifically *Bifidobacterium* and *Lactobacillales*, were significantly higher in the eggshell membrane group compared to the control group at 4 weeks (* *p* < 0.01) and 8 weeks (*p* < 0.05) (Figure 8).

A risk of bias assessment was conducted in accordance with the Cochrane Handbook (RoB2). The results are summarized in Appendix A, covering domains such as randomization, deviations from intended interventions, missing data, outcome measurement, and reporting. A CONSORT-style flow diagram summarizing the participant progression throughout the study is presented in Appendix A. Additionally, a completed CONSORT checklist and a table comparing prognostic and demographic characteristics between groups are included in Appendix A and Appendix A, respectively. Post hoc power analyses and effect sizes were calculated using G*Power 3.1 to evaluate the statistical validity of the findings (see Appendix A). The trial was prospectively registered in the UMIN Clinical Trials Registry (UMIN-CTR, No. 000057589).

## 3. Discussion

ESM is a complex biomaterial composed of more than 700 distinct proteins. In the industrially processed ESM powder used in this study, LYZ accounted for approximately 50% of the total protein content. When tritium-labeled ESM (^3^H-ESM), synthesized by neutron irradiation in vacuum-sealed quartz tubes in the presence of lithium carbonate, was orally administered to mice, its components were absorbed through the digestive tract and detected not only in the duodenum, small intestine, and colon but also in peripheral organs such as the skin, lungs, liver, and kidney via systemic circulation. (Distribution to other organs will be discussed in a separate publication.). Continuous intake of feed containing ESM in rodents has been reported to improve liver fibrosis and suppress inflammation in ulcerative colitis models [39,40]. However, such nutritional approaches make it difficult to compare dosage and timing with human supplement intake. Therefore, in this study, we adopted a method similar to the supplement administration protocol previously validated in a bleomycin-induced pulmonary fibrosis model [19] and calculated the appropriate dosage for mice by converting the standard human single-dose intake of ESM based on body weight.

In a DSS-induced colitis mouse model, pre-administration of ESM or LYZ for one week, followed by continuous intake, significantly suppressed DAI scores. Several previous studies have demonstrated the beneficial effects of LYZ supplementation in DSS-induced colitis models [41,42], and our findings are consistent with these reports. Therefore, it is plausible that the observed protective effect of ESM in this model can be partially attributed to its high LYZ content. Taken together, these results strongly suggest that LYZ is one of the key functional components responsible for the anti-colitic effects of ESM. However, ESM also contains a variety of other bioactive antimicrobial proteins, and we have recently shown that combining LYZ and ovotransferrin (OT) at their natural ratio in ESM synergistically enhances the secretion of decorin in lung fibroblasts [19]. Decorin is an extracellular proteoglycan that exerts intrinsic antifibrotic effects by capturing TGF-β in the extracellular space [43,44]. Whether similar synergistic effects contribute to the modulation of intestinal inflammation remains to be elucidated. Thus, the potential involvement of additional ESM components beyond LYZ warrants further investigation in future studies.

Although LYZ is an important component, the overall therapeutic effect of ESM is likely attributable to the combined actions of its many proteins. ESM is composed of more than 700 proteins, and even those present at levels exceeding 1% are expected to have multiple functions. Moreover, ESM is fibrous, and as reported in our 2011 study [45], hydrolyzed ESM can self-assemble into a mesh-like structure, as demonstrated by AFM imaging. Interestingly, LYZ is known to serve as a scaffold in eggshell formation. Therefore, fragments of ESM fibers or intestinal ECM may self-assemble around LYZ and provide mechanical stimulation to cells, thereby activating cellular responses such as mitochondrial activity and HSP induction. Highly cross-linked by lysyl oxidase and possessing unique self-assembly properties (including potential phase separation), ESM represents a distinctive fibrous complex capable of exerting mechanostimulatory and mild stress effects. These aspects warrant further investigation in future studies.

Dynamic nature of the gut microbiota is increasingly recognized as critical to host health. In this study, a double-blind, placebo-controlled intervention study in healthy adults demonstrated that twice-daily supplementation with ESM improved the balance of the gut microbiota, accompanied by an increase in beneficial bacteria such as *Bifidobacterium* and members of the order *Lactobacillales*. Interestingly, early colonization of *Bifidobacterium* in the fetal and neonatal gut has been reported to play a crucial role in the development of the gut microbiota and in promoting infant health [46]. Additionally, ESM and LYZ intake tended to restore the DSS-induced loss of gut microbiota diversity and promoted an increase in the relative abundance of *Lactobacillus*. These changes were consistent with results from a human supplementation study, in which increased microbiota diversity was observed as early as one week after initiating ESM intake, supporting the utility of the DSS colitis model as an effective system for evaluating ESM-induced improvements in the intestinal environment.

We have previously demonstrated that ESM stimulates fibroblasts to promote the production of “young” ECM components [37,45]. In this study, oral administration of ESM to healthy mice increased the type III to type I collagen expression ratio in the colon. Type III collagen expression in the mucous membrane tissue and submucosal tissues are shown in Figure 9. The increase in the type III/type I collagen gene expression ratio in the colon following eggshell membrane intake can be attributed to three main pathways: (1) absorbed components in the small intestine enter systemic circulation and act on intestinal cells, (2) undigested components reach the colon and directly affect epithelial or stromal cells, and (3) undigested components are metabolized by gut microbiota, and the resulting metabolites influence host cell activity. Pre-administration of ESM or LYZ before DSS exposure may protect colonic tissue from mucosal injury and reinforce basement membrane integrity. Lgr5-positive intestinal stem cells (ISCs) residing in the crypts of the intestinal epithelium maintain their stemness through interactions with surrounding supporting cells (e.g., Paneth cells) and the mechanical properties of the ECM [47]. Recent studies have demonstrated that substrate stiffness plays a critical role in regulating stem cell fate: increased ECM rigidity tends to promote differentiation, whereas softer matrices help preserve the undifferentiated state [48]. In this context, intestinal mucosa enriched in type III collagen—a more compliant and reticular ECM component—may provide a more favorable stem cell niche for maintaining Lgr5^+^ ISCs. In contrast, type I collagen, known for forming stiffer, fibrillar structures, is often associated with enhanced cellular differentiation and reduced stemness. Therefore, a higher type III/type I collagen ratio (Col3/Col1) in the intestinal ECM may contribute to a more permissive microenvironment for ISC maintenance and epithelial homeostasis. This insight holds potential implications for regenerative medicine, organoid culture optimization, and the mechanobiological understanding of epithelial stem cell regulation.

Notably, colonic epithelial cells are known to sense mechanical stress caused by fecal bulk and luminal distension via Piezo1 and Piezo2 mechanosensitive ion channels [49]. In response, they regulate water and electrolyte (Na^+^, Cl^−^) absorption through TRPV4 channels [50]. In this context, the ECM may influence the elasticity of the mucous membrane and submucosal tissue, thereby modulating the mechanical interaction between the tissue and bacteria. Systemically absorbed ESM, after digestion and circulation, likely affects ECM production in the colon. Regarding the modulation of the gut microbiota, antimicrobial peptides (AMPs) such as LYZ—which are abundant in ESM—are generally cationic and tend to bind to the negatively charged bacterial surface [51,52]. The surface of intestinal bacteria is generally negatively charged due to the presence of anionic components such as lipopolysaccharides and teichoic acids [53]. Zeta potential measurements for example −8.0 to −13.8 mV, reflecting their electrostatic properties and influencing their adhesion potential to host tissues [53]. It is therefore conceivable that undigested or partially digested ESM excreted in feces may bind more bacteria than control feces, serving as a potential trigger for microbiota compositional changes. These findings suggest that the effects of ESM on colonic health are mediated not only through direct tissue remodeling but also through interactions involving ESM-containing feces.

The primary limitation is the small sample size in both the animal experiments and the human trial, which may have restricted the statistical power. On the other hand, the strengths include: (i) proteomic characterization of the ESM protein components prior to intervention, thereby clarifying the major functional constituents for readers; (ii) the use of a unique labeling method that enabled tritium labeling of insoluble composite proteins, allowing oral administration to mice in a manner comparable to human supplementation; and (iii) demonstration that, beyond the transient direct effects on gut microbiota as observed in fecal absorption peaks, absorbed components were distributed via the bloodstream to the colon and other organs. These findings complement our previous reports on the effects of ESM supplementation in the skin and lung, while bridging the knowledge gap on its intestinal effects in the context of IBD by showing consistent microbiota changes in both mice and humans.

## 4. Materials and Methods

### 4.1. Eggshell Membrane Powder for Oral Administration

Eggshell membrane, a by-product of the food industry, is water-insoluble and bulky, making it difficult to formulate into tablets for human supplementation. In this study, we used eggshell membrane powder processed into a food-grade fine powder by Almado Inc. (Tokyo, Japan) [18,19,23] as an oral supplement material for both humans and mice.

### 4.2. Proteomics Analysis

To obtain information on the functional protein components in the eggshell membrane powder (Almado Inc.) used in this study, proteomics analysis based on data-independent acquisition (DIA) mass spectrometry provided by Promega Corporation was performed [54]. The eggshell membrane samples were pretreated according to the following procedure prior to proteomics analysis. First, acetonitrile containing trifluoroacetic acid (TFA) was added, and the samples were subjected to ultrasonic treatment to denature and solubilize the proteins and inactivate endogenous enzymes. After centrifugation, sodium dodecanoate (SDoD) was added to the precipitate, and the proteins were dissolved using a closed ultrasonic disruption device. The concentration of the protein solution was measured using a BCA assay and adjusted to 1 μg/μL. Next, DTT was added to reduce disulfide bonds, and the mixture was incubated at 50 °C. Iodacetamide (IAA) was then added to alkylate cysteine residues, and the mixture was reacted at room temperature under light-protected conditions for 30 min. To remove unreacted IAA, cysteine was added, and the mixture was incubated at room temperature for 10 min. To enhance the enzymatic digestion efficiency of the protein, ammonium bicarbonate was added, followed by Lys-C (400 ng) and trypsin (400 ng), and the mixture was incubated at 37 °C overnight. After enzymatic digestion, TFA was added to acidify the solution, and centrifugation was performed to precipitate and remove SDoD. The supernatant was collected. The resulting peptide solution was desalted using a C18 spin column and dried to a dry residue using a centrifugal evaporator. After drying, the solution was redissolved in a mixture of acetonitrile and formic acid and homogenized using a closed ultrasonic homogenizer. The peptide concentration was measured again using the BCA assay, adjusted to 100 ng/μL, and subjected to LC-MS/MS analysis. The prepared peptides were analyzed by nano liquid chromatography-mass spectrometry (nanoLC-MS/MS) using an UltiMate 3000 RSLCnano LC system (Thermo Fisher Scientific K.K., Tokyo, Japan). 500 ng of peptides were injected, and separation was performed using a CAPCELL CORE MP column (Osaka Soda Co., Ltd., Osaka, Japan) maintained at 50 °C. The mobile phase consisted of water containing 0.1% formic acid (solvent A) and 80% acetonitrile containing 0.1% formic acid (solvent B). The obtained MS data were analyzed using Scaffold DIA 2.2.0 (Proteome Software, Inc., Portland, OR, USA) for protein and peptide identification and quantification. Quantification was based on the MS peak intensity of peptides, and relative abundance was calculated. The protein sequence database used was Gallus gallus (UP000000539) obtained from UniProt release 2020_05 (UniProt consortium, EMBL-EBI, Cambridge, UK). The spectral library was created using Prosit (https://www.proteomicsdb.org/prosit/ (accessed on 10 November 2020)) based on the aforementioned sequence database.

### 4.3. Tritium Labeling of Eggshell Membrane (ESM)

Based on our previously reported method, a mixture of ESM powder (Almado Inc.) and lithium carbonate (Li_2_CO_3_) at a weight ratio of 65:35 was vacuum-sealed in a custom-made quartz glass ampoule (Kiriyama Glass Works Co., Tokyo, Japan). Tritium labeling was performed via the ^6^Li(n,α)^3^H nuclear reaction, which has been employed for radiolabeling of natural products that are difficult to chemically synthesize and for investigating their tissue distribution in vivo. The sealed ampoule was irradiated under mild conditions (Pn-2 reactor, 1 MW, 70 min) at the Kyoto University Institute for Integrated Radiation and Nuclear Science (KURNS) through a collaborative research program. Following irradiation, the labeled ESM, being insoluble in water, was repeatedly washed with ultrapure water to remove residual lithium carbonate and then dried.

### 4.4. In Vivo Distribution of Tritium-Labeled Eggshell Membrane Powder in Mice

Female C57BL/6J Kwl mice (6 weeks old) were acclimated for one week under controlled environmental conditions (The light-dark cycle was set from 8 a.m. to 8 p.m. as the light period and the remaining hours as the dark period, and the rearing room was maintained at a temperature of 19–23 °C and a humidity of 35–45%). Each mouse was housed individually in a metabolic cage (Type 3600M021, TECNIPLAST JAPAN Co., Ltd., Tokyo, Japan) with ad libitum access to food and ultrapure water. Prior to administration, mice (average body weight: 17.0 ± 0.6 g) were fasted for 3–6 h. A single oral dose of tritium-labeled eggshell membrane (5 mg [^3^H]ESM, 1.3 × 10^4^ Bq) suspended in MediGel^®^ Sucralose (ClearH2O, Portland, ME, USA) was administered. At 0.5, 2, 6, 24, 48, and 72 h post-administration, mice were euthanized by overdose of isoflurane inhalation, and blood and tissue samples were collected. Samples were snap-frozen and stored at −80 °C until analysis. For radioactivity measurement, tissues were solubilized using SOLVABLE™ (PerkinElmer Japan G.K., Kanagawa, Japan), mixed with scintillation cocktail (Ultima Gold™, PerkinElmer), and measured using a liquid scintillation counter (Tri-Carb 3100TR, PerkinElmer) after 1 h of stabilization.

### 4.5. Oral Administration of Eggshell Membrane in DSS-Induced Colitis Model

Female C57BL/6J mice (6 weeks old, body weight 16.3 ± 0.6 g) were used in the study. Upon arrival, the animals were housed individually in cages with free access to food (CE-2, CLEA Japan, Inc., Tokyo, Japan) and water. The animal room was maintained at a temperature of 19–23 °C and humidity of 35–45%, with a 12 h light/dark cycle (lights on from 8:00 to 20:00). Body weight and water intake were measured daily to monitor animal health. To induce colitis, 2.0% (*w*/*v*) dextran sulfate sodium (DSS; Sigma-Aldrich, Tokyo, Japan) was added to the drinking water. ESM (7.4 mg/kg/dose) and LYZ (Sigma-Aldrich) (11.98 mg/kg/dose) were suspended in MediGel^®^ (ClearH2O, Portland, ME, USA) and orally administered. Mice were divided into four groups: (1) control (DSS-, MediGel only), (2) DSS group 1 (DSS+, MediGel only), (3) DSS group 2 (DSS+, ESM-MediGel), and (4) DSS group 3 (DSS+, LYZ-MediGel). Following one week of pre-treatment with the supplements, DSS administration was initiated via drinking water. During the DSS treatment period, supplements were continuously administered once daily. On days 3 and 5 after DSS administration, mice were sacrificed, and colons were collected for histopathological analysis (n = 4 per group).

Throughout DSS administration, body weight, stool consistency, and the presence of blood in the stool were recorded daily to calculate the disease activity index (DAI). DAI was determined as the sum of scores from the following three parameters:Body weight loss (relative to initial weight): 0 = no loss; 1 = 1–10%; 2 = 10–20%; 3 = >20%.Stool consistency: 0 = normal; 1 = soft; 2 = diarrhea; 3 = watery diarrhea.Fecal blood: 0 = none; 1 = partial bleeding; 2 = extensive bleeding; 3 = blood around the anus.

#### The Average Daily DAI Score for Each Group Was Calculated

The excised mouse intestine was carefully spread on graph paper to minimize curvature and photographed together with a ruler using a Canon EOS 6D digital camera, Canon Inc., Tokyo, Japan mounted on a fixed stand. The acquired images were analyzed using ImageJ2 (version 2.16.0/1.54p). A known distance was calibrated with the ruler, and colon length was determined by drawing a straight line from the anal end to the cecum. Data are expressed as relative colon length normalized to the average control colon length. Error bars represent SD (n = 4). ** *p* < 0.01, * *p* < 0.05.

For histological evaluation, the excised colons were fixed in 10% neutral-buffered formalin and sent to Genostaff Co., Ltd. (Tokyo, Japan) for paraffin embedding, sectioning, hematoxylin and eosin (HE) staining, and digital microscopy imaging. Histological grading of colitis was performed according to the method of Cooper et al. [28]. Lesions were evaluated for mucosal damage (D) and extension (E), each scored from 0 to 4. Damage was graded as follows: 0, none; 1, loss of the basal one-third of the crypt; 2, loss of the basal two-thirds; 3, complete crypt loss with intact surface epithelium; and 4, loss of both crypt and surface epithelium (erosion). Extension was scored as 0, none; 1, focal; 2, one-third of the intestine; 3, two-thirds; and 4, entire intestine. The histological index (HI) was calculated as the sum of D and E. For each animal, the HI was expressed as the mean of the scores obtained from two sections of the same animal. All histological evaluations were performed in a blinded manner. Statistical comparisons among groups were performed using one-way ANOVA followed by Tukey’s post hoc test, with significance set at *p* < 0.05.

For ratio of sirius Red–positive staining in submucosa relative to mucosa, regions of interest (ROIs) were manually delineated for the mucosa and submucosa under blinded conditions. Quantification of Picrosirius Red–positive regions was restricted using the Limit to threshold option in ImageJ. For each ROI, mean saturation values were calculated by dividing the integrated density (IntDen) of thresholded pixels by the corresponding positive area (Area). The submucosa/mucosa ratio of mean saturation was then determined for each sample. Data are shown for Control, DSS, ESM + DSS, and LYZ + DSS groups (n = 4 per group) and are expressed as mean ± SD. Statistical analysis was performed using one-way ANOVA followed by Tukey’s multiple comparison test; *p* < 0.05 was considered significant.

For gut microbiota analysis, feces collected from the colon on day 5 of DSS administration (2–3 pellets or >20 mg) were immediately transferred to sterilized tubes and stored at –80 °C until shipment to the ICLAS Monitoring Center, Central Institute for Experimental Animals (CIEA), Kawasaki, Japan. Fecal samples were suspended in GTC solution (4 M guanidine thiocyanate, 100 mM Tris-HCl pH 9.0, 40 mM EDTA pH 8.0), and DNA was extracted using a bead-beating and phenol-chloroform method. The bacterial 16S rDNA was subjected to T-RFLP analysis. Fragment analysis was performed using an ABI 310 Genetic Analyzer and GeneScan software 3.7 (Applied Biosystems, Foster, CA, USA). Operational taxonomic units (OTUs) were identified based on fragment lengths and assigned to bacterial groups using a mouse intestinal microbiota database [38,55]. The relative abundance of estimated bacterial groups, based on OTU peak area ratios, was visualized using pie charts.

### 4.6. Evaluation of the Effects of Oral Administration of Eggshell Membrane Fine Powder and Lysozyme on the Expression of Genes Related to the Intestinal ECM in Mice

To evaluate the effects of orally administered ESM fine powder and LYZ on ECM-related gene expression in the mouse intestine, male C57BL/6J mice (8 weeks old, body weight 21.5 ± 0.8 g) were used. ESM (7.4 mg/kg) or LYZ (5.99 mg/kg) was suspended in MediGel^®^ (ClearH2O, Portland, ME, USA) and administered once daily for 2 weeks. At the end of the treatment period, mice were anesthetized with isoflurane (FUJIFILM Wako Pure Chemical Corp., Osaka, Japan), followed by abdominal blood collection and dissection. The intestine was excised, immediately snap-frozen in liquid nitrogen, and stored at −80 °C until analysis. Total RNA was extracted from frozen intestinal tissue using TRI Reagent^®^ (Cosmo Bio Co., Ltd., Tokyo, Japan). Reverse transcription and subsequent real-time PCR reactions were performed using the GoTaq^®^ Probe 2-Step RT-qPCR System (Promega, Madison, WI, USA), and amplification was conducted on a Thermal Cycler Dice^®^ Real Time System II (Takara Bio, Shiga, Japan). Relative gene expression levels were quantified using the standard curve method, normalized to the housekeeping gene GAPDH. The expression ratio of Col3a1 to Col1a1 was also calculated for each group. Primer sequences used were as follows: gapdh-Fw: GCCAGCCTCGTCCCGTAG; gapdh-Rv: AATCTCCACTTTGCCACTGCA; Col1a1-Fw: CACTGCCCTCCTGACGCATG; Col1a1-Rv: TCAAGCATACCTCGGGTTTCCA; Col3a1-Fw: GTGATGAGGAGCCACTAGACTG; Col3a1-Rv: AGGAAGCACAGGAGCAGGT.

### 4.7. Human Study

This study was conducted concurrently with our previous report, which evaluated functional outcomes including skin elasticity, respiratory function, and physical performance [18]. Twenty healthy, non-smoking male and female volunteers (age range: 22–68 years; mean ± SD: 37.1 ± 12.6 years) who agreed to participate in the study were enrolled. Inclusion criteria required that participants had no notable current medical conditions at the time of enrollment. Exclusion criteria included a history of food allergies, cardiovascular disease, hypertension, thyroid disorders, medical advice prohibiting exercise, and regular use of dietary supplements. The study was conducted at Tokyo University of Agriculture and Technology over 8 weeks from September to November 2015, with participant recruitment carried out during a two-week period in August 2015. Participants were randomly assigned to two groups: the eggshell membrane (ESM) group (n = 10, mean age: 39.8 ± 15.6 years) and the control group (n = 10, mean age: 39.1 ± 14.6 years). The human study was conducted in a double-blind manner. Participants ingested four tablets in the morning and four in the evening (eight tablets daily) for 8 weeks. Each tablet weighed 493 mg and measured 10 mm in diameter. The ESM group received tablets containing 46% eggshell membrane powder (Almado Inc., Tokyo, Japan) (7.4 mg/kg/dose), 39% lactose, and 15% other functional ingredients, including chicken breast extract (anserine and carnosine), rice germ extract, Jew’s mallow powder, dried royal jelly, pig-derived elastin peptide, chicken cartilage extract, acerola powder, canola hardened oil, Japanese apricot extract, corn germ extract ceramide inclusion, dextrin, vitamin E-containing plant oil, proteoglycan-containing salmon nasal cartilage extract, corn protein, hyaluronan, cyclic oligosaccharide, lycopene-containing tomato powder, extracted carotene, eggshell calcium, vitamin C, biotin, and vitamin B2. The sugar coating included granulated sugar, syrup containing rare sugars, eggshell calcium, shellac, gelatin, Arabic gum, carnauba wax, and gardenia pigment. The control group received identical tablets in appearance and composition, but without the inclusion of ESM. Participants were instructed to refrain from taking other dietary supplements or undergoing body aesthetic treatments during the study period. The supplement used in this study was considered safe. In the event of adverse effects related to the supplement or study procedures, participants were advised to seek prompt medical attention, and appropriate treatment and compensation were guaranteed if necessary.

### 4.8. Fecal Sample Collection and Gut Microbiota Analysis by T-RFLP

Fecal samples were collected from the participants at six time points: before supplementation (pre), and at 1, 2, 4, 8 weeks after the start of supplementation, using a fecal collection kit provided by TechnoSuruga Laboratory Co., Ltd., Shizuoka, Japan. The samples were analyzed by TechnoSuruga Laboratory using the terminal restriction fragment length polymorphism (T-RFLP) method (Nagashima method) to assess the composition of the gut microbiota. Based on the operational taxonomic units (OTUs), the relative abundance of the following bacterial groups was estimated: *Bifidobacterium*, *Lactobacillales*, *Bacteroides*, *Prevotella*, *Clostridium cluster IV*, *Clostridium subcluster XIVa*, *Clostridium cluster XI*, *Clostridium cluster XVIII*, and *others*.

Enterotype classification was determined based on the relative abundance of Bacteroides. Data analysis was performed on 17 participants (10 in the control group and 7 in the ESM group) who completed all sampling time points and whose baseline (pre) microbiota composition was classified as enterotype 1, characterized by *Bacteroides* dominance, which is considered a typical pattern among adults, including Japanese [56].

### 4.9. Assessment of α-Diversity of Gut Microbiota

The α-diversity of the gut microbiota was assessed using the Shannon-Wiener Diversity Index (H) as described by Shannon (C.E.), calculated using the following formula: H = −Σ (pi × ln pi)
where Σ represents summation, and pi is the proportion of species i, calculated as the number of individuals of species i divided by the total number of all observed species. The Shannon index was calculated for each participant at each time point based on the relative abundance of 10 major bacterial orders/genera. Temporal changes during the supplementation period and intergroup comparisons were assessed.

### 4.10. Statistical Analysis

Data are expressed as mean ± standard deviation (SD). For multiple group comparisons (Figure 2), one-way ANOVA was used to determine statistical significance. For comparisons within the same group before and after supplementation, paired Student’s *t*-tests were used. For comparisons between the ESM and control groups at corresponding time points, unpaired Student’s *t*-tests were applied. A *p*-value of less than 0.05 was considered statistically significant.

## 5. Conclusions

Proteomic analysis of the protein components of ESM, which is known to promote wound healing, rejuvenate the ECM, and attenuate fibrosis in the liver and lung, revealed that approximately 50% consisted of LYZ, followed by lysyl oxidase. Oral administration of tritium-labeled ESM demonstrated its digestion, absorption, and systemic distribution, providing a rationale for the multi-organ effects of ESM. In the context of IBD, we addressed the knowledge gap regarding the in vivo effects of ESM in humans by focusing on fecal microbiota, which can be assessed noninvasively. Our results showed that ESM intake improved microbial diversity, and in a double-blind, placebo-controlled trial in healthy adults, significant increases in beneficial bacteria were observed at weeks 4 and 8. Taken together, these findings suggest that ESM supplementation holds promise as a novel preventive approach for intestinal diseases with unmet medical needs.

## Figures and Tables

**Figure 1 ijms-26-09102-f001:**
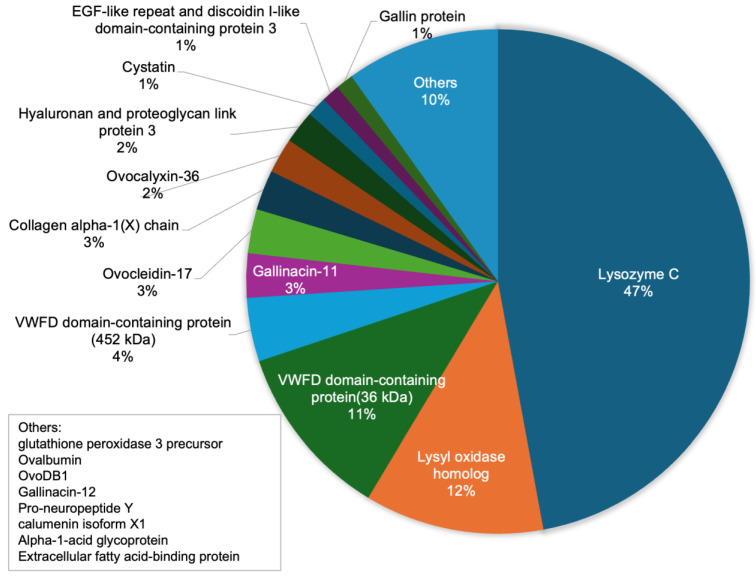
Major proteins identified in the powdered eggshell membrane sample. Proteins accounting for more than 1% of the total identified protein content are shown. Notable components include lysozyme, lysyl oxidase homolog, type X collagen, and other antimicrobial or inhibitory proteins, suggesting the multifunctional properties of the eggshell membrane.

**Figure 2 ijms-26-09102-f002:**
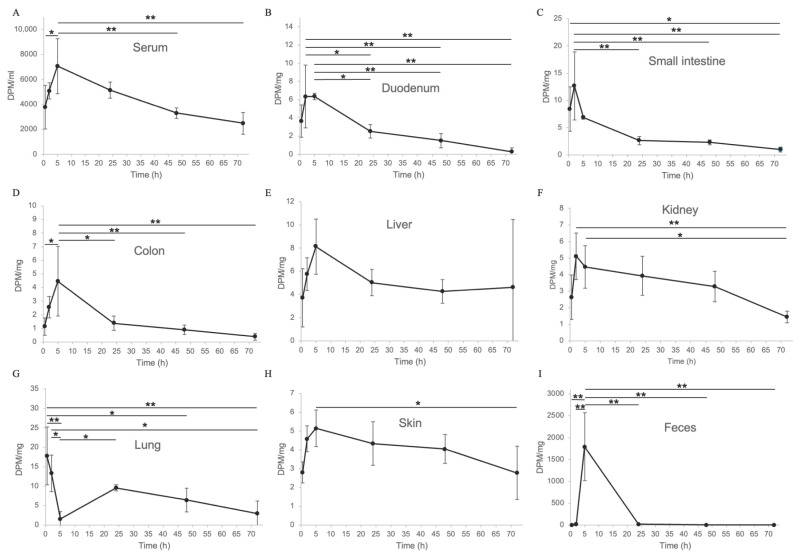
Tissue distribution of tritium-labeled eggshell membrane components in mice after oral administration. Radioactivity concentrations were measured in blood (**A**), organs (duodenum (**B**), small intestine (**C**), large intestine (**D**), liver (**E**), kidney (**F**), lung (**G**), skin (**H**), and rectal feces (**I**) at 0.5, 2, 5, 24, 48, and 72 h post-administration. Blood radioactivity peaked at 5 h, reflecting systemic absorption and distribution of eggshell membrane (ESM)-derived components. Data represent the mean of n = 3 (0.5 and 2 h) or n = 4 (5, 24, 48, and 72 h) mice, with error bars indicating standard deviation (SD). Note that duodenum and small intestine data at 5 h are based on n = 3. For multiple group comparisons, one-way ANOVA was used to determine statistical significance. * *p* < 0.05, ** *p* < 0.01.

**Figure 3 ijms-26-09102-f003:**
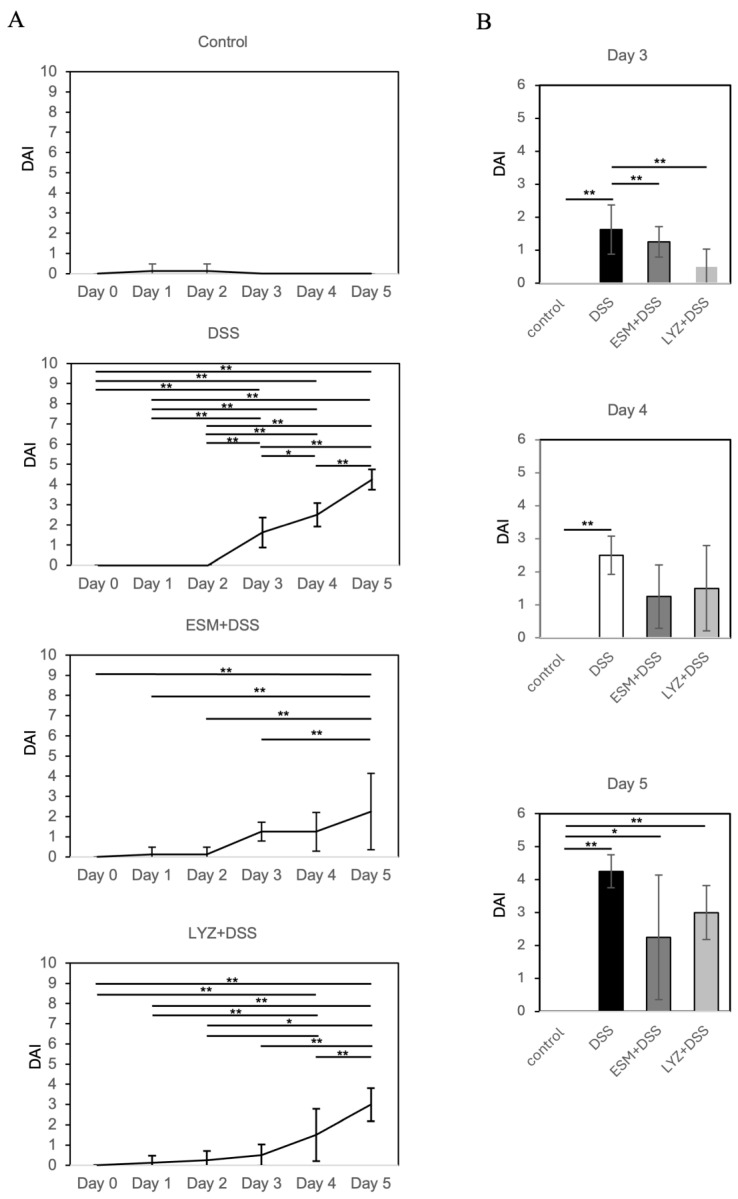
Effects of eggshell membrane (ESM) and its major component lysozyme (LYZ) on DSS-induced colitis, evaluated by disease activity index (DAI). Days indicate the period after DSS administration. Experimental groups included control, DSS, ESM + DSS, and LYZ + DSS. On day 3, half of the mice (n = 4) were sacrificed for histological analysis, while the remaining mice continued to receive DSS for an additional 2 days (n = 4). The control group without DSS maintained a DAI score of 0. The time course of DAI for each group is shown in (**A**), and differences among the four groups at days 3, 4, and 5 are shown in (**B**). Statistical analysis was performed using one-way ANOVA followed by Tukey’s multiple comparison test, with *p* < 0.05 considered significant. Error bars indicate standard deviation (SD). * *p* < 0.05, ** *p* < 0.01.

**Figure 4 ijms-26-09102-f004:**
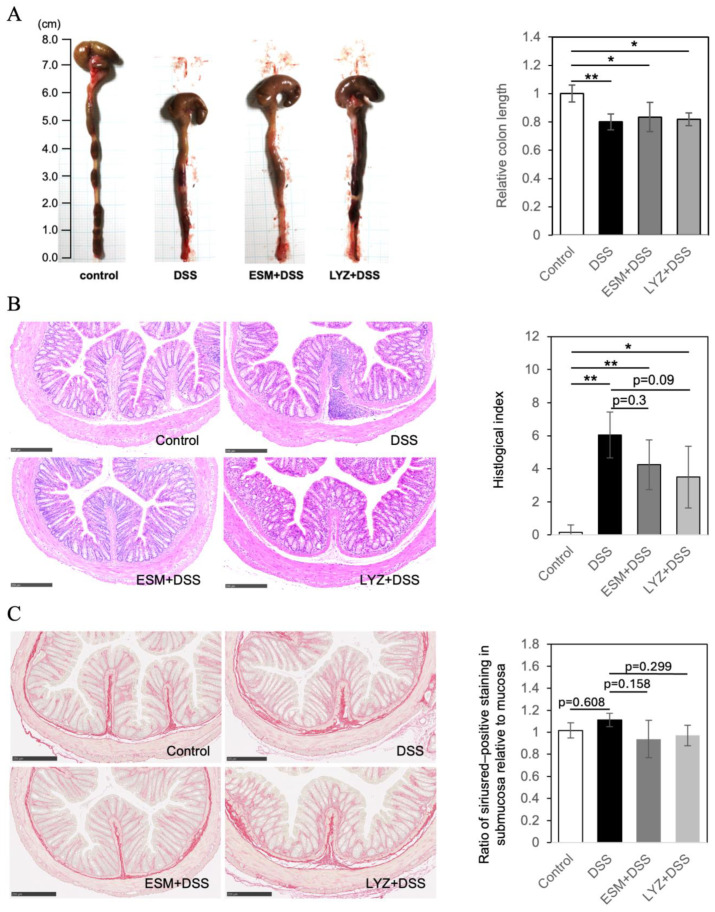
Typical macroscopic appearance (**A**) and histological images (**B**,**C**) of control mice, DSS-induced colitis mice, and mice pretreated with ESM or LYZ. Colon length is shown in (**A**). Colitis severity was graded according to the method of Cooper et al. [28]; using hematoxylin and eosin (HE) -stained sections the histological index was calculated as the sum of the mucosal damage score and the extension of the lesion (**B**). The ratio of Sirius Red–positive staining in the submucosa relative to the mucosa is shown in (**C**). Statistical analysis was performed using one-way ANOVA followed by Tukey’s multiple comparison test, with *p* < 0.05 considered significant. Scale bars: 250 μm in (**B**,**C**). * *p* < 0.05, ** *p* < 0.01.

**Figure 5 ijms-26-09102-f005:**
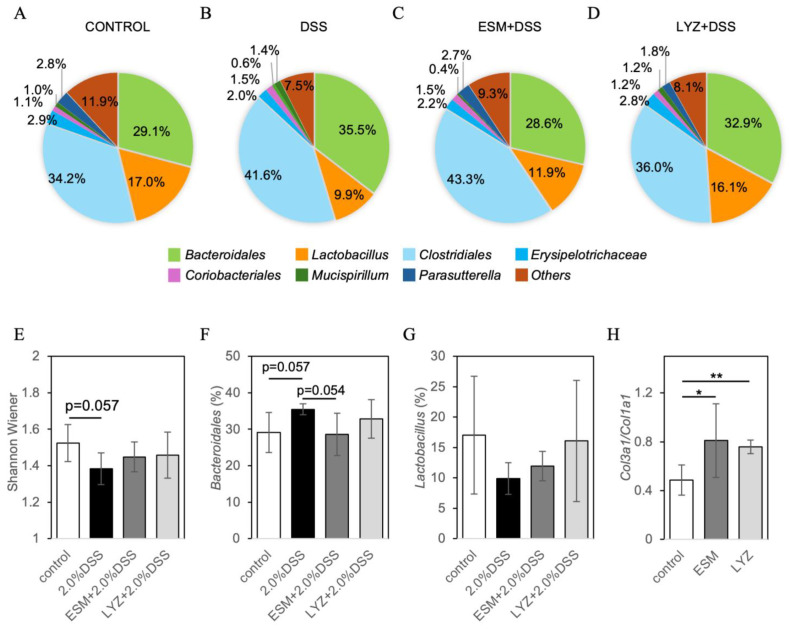
Effects of eggshell membrane (ESM) and lysozyme (LYZ) intake on gut microbiota composition and collagen gene expression in DSS-induced colitis model mice. (**A**–**D**) Pie charts showing estimated bacterial group composition based on T-RFLP peak area ratios derived from OTU assignments: control (**A**), DSS-only (**B**), DSS + ESM (**C**), and DSS + LYZ (**D**). (**E**) Shannon-Wiener diversity index calculated from OTU profiles. The DSS group showed a decreasing trend in microbial diversity (*p* = 0.057), which was attenuated by ESM and LYZ intake. (**F**,**G**) Relative abundance of *Bacteroidales* (**F**) and *Lactobacillus* (**G**). The DSS group showed an increase in *Bacteroidales* (*p* = 0.057), while the DSS + ESM group maintained levels comparable to the control. *Lactobacillus* abundance showed a similar trend to the diversity index. n = 4 for (**A**–**G**), except DSS group (n = 3). (**H**) Relative Col3a1/Col1a1 gene expression ratio in the rectum after two weeks of ESM or LYZ intake in wild-type mice (n = 4). A significant increase was observed in both treatment groups. Error bars indicate SD. * *p* < 0.05, ** *p* < 0.01.

**Figure 6 ijms-26-09102-f006:**
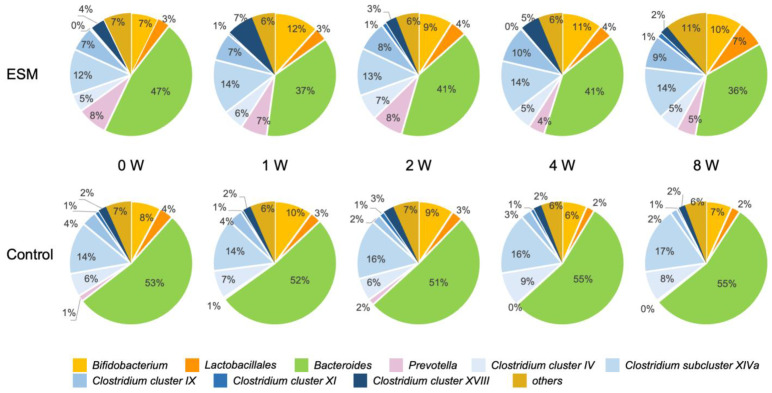
Time-dependent effects of eggshell membrane intake in healthy individuals: Gut microbiota profiling by T-RFLP analysis. Control group (n = 10), eggshell membrane (ESM) group (n = 7).

**Figure 7 ijms-26-09102-f007:**
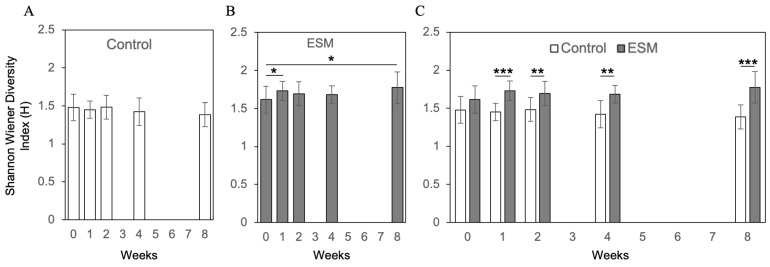
Temporal changes in gut microbiota α-diversity as measured by the Shannon-Wiener diversity index (H). (**A**) Control group without eggshell membrane intake (n = 10), (**B**) Eggshell membrane intake group (n = 7), and (**C**) Comparison between groups. * *p* < 0.05, ** *p* < 0.01, *** *p* < 0.001.

**Figure 8 ijms-26-09102-f008:**
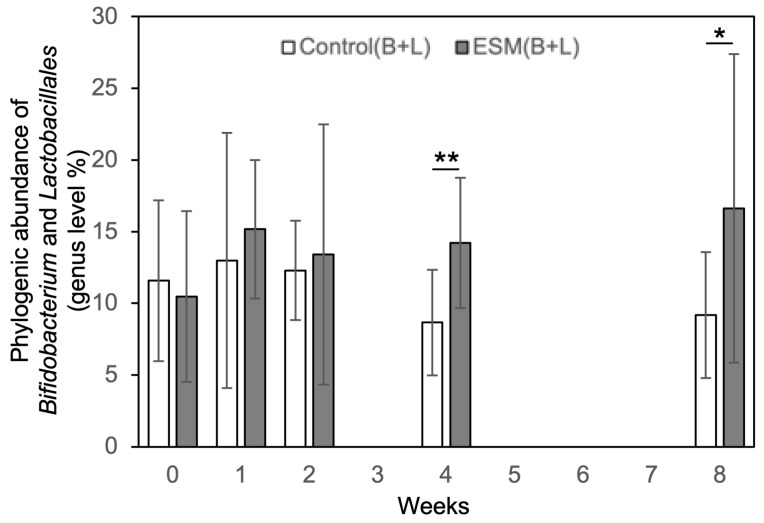
Temporal changes in the proportion of beneficial bacteria (*Bifidobacterium* and *Lactobacillales*) following supplement intake. At weeks 4 and 8, the eggshell membrane group showed a significantly higher proportion of beneficial bacteria compared to the control group. Control group: n = 10; eggshell membrane group: n = 8. * *p* < 0.05, ** *p* < 0.01.

**Figure 9 ijms-26-09102-f009:**
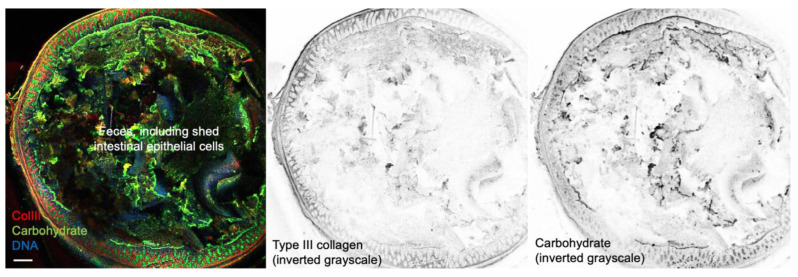
Type III collagen is expressed in the mucosal and submucosal tissues of the colon. Shown is a confocal image of a 100 μm-thick cross sectional vibratome slice (Leica VT1200S, Leica Microsystems K.K., Tokyo, Japan) of formalin-fixed, agarose-embedded mouse colon containing feces. The sample was stained for DNA (Hoechst 33342, blue), carbohydrates [wheat germ agglutinin (WGA), which binds strongly to sialic acid and *N*-acetylglucosamine, conjugated to Alexa Fluor 488, green], and polyclonal anti-type III collagen antibody (LB-1387, Cosmo Bio Co., Ltd., Tokyo, Japan) followed by anti-rabbit Alexa555 conjugated secondary antibody. Scale bar: 200 μm.

## Data Availability

All data generated or analyzed during this study are included in this article.

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
