# Peer review of "Pharmacokinetic Profiling Using 3H-Labeled Eggshell Membrane and Effects of Eggshell Membrane and Lysozyme Oral Supplementation on DSS-Induced Colitis and Human Gut Microbiota"

_ijms, 2025, doi:10.3390/ijms26189102_

Round 1
Reviewer 1 Report
Comments and Suggestions for Authors
- In the Methods section of “Oral Administration of Eggshell Membrane in DSS-Induced Colitis Model,” it states: “On days 3 and 5 after DSS administration, mice were sacrificed, and colons were collected for histopathological analysis.” Please clarify the rationale for choosing days 3 and 5 post-DSS administration for sacrificing the mice.
- For Figure 3, please provide a plot showing the changes in DAI values over time.
- For Figure 4, please include a statistical analysis of colon length (e.g., bar graph) and indicate statistical significance.
- For Figure 5, please include statistical analysis of the histopathological scores in a graphical format.
- For Figure 10, please provide images for the individual fluorescence channels (e.g., green, red).
Author Response
We sincerely appreciate the Editor and Reviewers for their insightful comments and valuable suggestions, which have greatly helped us to improve our manuscript. We have carefully addressed each comment, revised the manuscript accordingly, and highlighted the changes in the revised version. We believe that these revisions have enhanced both the scientific rigor and readability of the paper. Detailed point-by-point responses are provided below.
Comments 1: In the Methods section of “Oral Administration of Eggshell Membrane in DSS-Induced Colitis Model,” it states: “On days 3 and 5 after DSS administration, mice were sacrificed, and colons were collected for histopathological analysis.” Please clarify the rationale for choosing days 3 and 5 post-DSS administration for sacrificing the mice.
Response to Comments 1: We selected days 3 and 5 after DSS administration as endpoints for histological analysis for the following reasons. Since the effect of ESM is relatively mild, it is important to evaluate disease progression without allowing the pathology to become too severe. DSS-induced colitis is an acute model, and the 5-day administration period was chosen based on previous reports. In our preliminary experiments, DAI scores were already evident at day 3; therefore, we included this earlier time point as an endpoint to capture the initial pathological changes.
Comments 2: For Figure 3, please provide a plot showing the changes in DAI values over time.
Response to Comments 2:
As suggested, we have added a plot showing the time course of DAI values for each of the four groups (Figure 3). In addition, group comparisons at each time point are presented in a separate graph.
Comments 3: For Figure 4, please include a statistical analysis of colon length (e.g., bar graph) and indicate statistical significance.
Response to Comments 3:
As requested, we measured colon length and added a bar graph with statistical analysis to indicate significance (Figure 4A).
Comments 4: For Figure 5, please include statistical analysis of the histopathological scores in a graphical format.
Response to Comments 4:
Colitis severity was graded using the method of Cooper et al. and the histopathological scores were plotted with statistical analysis (Figure 4B).
Comments 5: For Figure 10, please provide images for the individual fluorescence channels (e.g., green, red).
Response to Comments 5:
We have added the separated images for the red, green, and blue fluorescence channels. As several figures were consolidated, Figure 10 has been renumbered as Figure 9.
Reviewer 2 Report
Comments and Suggestions for Authors
I consider that the manuscript titled: “Membrane and Effects of Eggshell Membrane and Lysozyme 3 Oral Supplementation on DSS-Induced Colitis and Human 4 Gut Microbiota” is good, but it needs to be restarted, mainly in discussion.
Based on a thorough review of the manuscript, here are the comments and suggestions for a revision, following the guidelines of the International Journal of Molecular Sciences.
Introduction
- The paragraph from lines 56 to 65 is cited by 24 references, I consider that there are a lot of references for such a small paragraph.
- The paragraph from lines 94 to 100, the objectives are mentioned and it’s are well-defined. I also suggest mentioning the following objective: to provide a knowledge gap regarding the in vivo effects of ESM in humans in the context of IBD.
Results
- Line 131: It is stated that “absorbed components reached peak blood concentrations at 5 hours". This is a key finding and I suggest that it should be discussed in the context of other absorbed components.
- Line 142: The explanation for “the early peak in the lungs”, attributed to bypassing the first-pass effect, is intriguing. I suggest a more detailed explanation or a diagram illustrating this proposed pathway would be a valuable addition to the discussion section.
- Line 192: Give a space after the period. And the authors correctly note that there were no statistically significant differences were observed in the macroscopic evaluation for colon length restoration. However, I consider it important to highlight this limitation and focus on the histological and molecular findings, which are more conclusive.
- Lines 234-237: The authors should discuss these as strong trends that, with a larger sample size, might become statistically significant.
Discussion
- Lines 319-321: The question of synergistic effects between LYZ and other ESM components is a good point for future research. I suggest that the authors should discuss this more thoroughly, emphasizing that while LYZ is a key component, ESM's full therapeutic effect is likely due to the combined action of its many proteins involved.
- I suggest that the discussion should also include a paragraph on the limitations of the study, such as the small sample sizes in both, the animal and human trials, which impacts the statistical power of the findings.
Author Response
We sincerely appreciate the Editor and Reviewers for their insightful comments and valuable suggestions, which have greatly helped us to improve our manuscript. We have carefully addressed each comment, revised the manuscript accordingly, and highlighted the changes in the revised version. We believe that these revisions have enhanced both the scientific rigor and readability of the paper. Detailed point-by-point responses are provided below.
Comments 1: Introduction 1
The paragraph from lines 56 to 65 is cited by 24 references, I consider that there are a lot of references for such a small paragraph.
Revised Response to Comment 1 (Introduction):
We have revised the paragraph and consolidated the citations, reducing the number of references in the Introduction from 46 to 25.
Comments 2: Introduction 2
The paragraph from lines 94 to 100, the objectives are mentioned and it’s are well-defined. I also suggest mentioning the following objective: to provide a knowledge gap regarding the in vivo effects of ESM in humans in the context of IBD.
Response to Comments 2 (Introduction):
We have revised the objectives as suggested and added the point regarding the knowledge gap on the in vivo effects of ESM in humans in the context of IBD.
Comments 3: Results 1
Line 131: It is stated that “absorbed components reached peak blood concentrations at 5 hours". This is a key finding and I suggest that it should be discussed in the context of other absorbed components.
Response to Comments 3-Results 1:
We agree that this is an important point. As shown in Results 1, eggshell membrane is a composite material. In the present study, tritium labeling was performed under neutron irradiation in the presence of lithium carbonate, which labels all protein-derived hydrogens indiscriminately. Therefore, it was not possible to distinguish or compare the kinetics of individual components in this experiment. To address this, we are planning further absorption studies in mice using tritium-labeled lysozyme and ovotransferrin, the major components of ESM. By comparing those results with the current data, we aim to clarify whether the observed in vivo kinetics are mainly attributable to these major components or to other constituents.
Comments 4: Results 2
Line 142: The explanation for “the early peak in the lungs”, attributed to bypassing the first-pass effect, is intriguing. I suggest a more detailed explanation or a diagram illustrating this proposed pathway would be a valuable addition to the discussion section.
Response to Commentsss 4:
We appreciate the reviewer’s insightful comment. In a previous methodological study we conducted in 2021, the number of mice examined was small; therefore, the conclusions cannot be considered definitive. In that experiment, the compounds were administered intragastrically by gavage, excluding oral mucosal absorption, and thus no first-pass effect should have occurred. In fact, no early absorption peak was observed in the lungs; rather, the peak appeared at 6 h, consistent with the serum peak. As this represents an important point for the design of nutrient and drug absorption studies, we plan to conduct further verification experiments and discuss this in more detail.
Comments 5: Results 3
Line 192: Give a space after the period. And the authors correctly note that there were no statistically significant differences were observed in the macroscopic evaluation for colon length restoration. However, I consider it important to highlight this limitation and focus on the histological and molecular findings, which are more conclusive.
Response to Comments 5 (Results 3):
We thank the reviewer for this valuable comment. In response, we have added the measurement graph to the Results section. The lack of a statistically significant effect on colon length restoration is likely due to the relatively small sample size. Moreover, the effect of ESM (and its major component, LYZ) is expected to be associated with rejuvenation of the intestinal ECM, leading to improved elasticity, rather than with macroscopic colon length restoration typically observed with other anti-colitis drugs. Sirius Red (SR) staining, which specifically distinguishes collagen fibers and is more suitable for fibrosis quantification than H&E or Masson’s trichrome, showed a tendency toward a lower submucosa/mucosa ratio in ESM- and LYZ-treated mice compared with DSS alone. This suggests attenuation of the DSS-induced increase in submucosal type I collagen deposition, which is known to stiffen the colon, impair peristalsis, and reduce water absorption. In addition, the mucosal staining pattern in the ESM and LYZ groups appeared finer and sharper compared with control or DSS. Our previous studies demonstrated that ESM promotes type III collagen, decorin, and MMP2 expression in dermal fibroblasts, thereby rejuvenating the tissue matrix. Since Sirius Red also stains type III collagen, it is possible that ESM and LYZ increased type III collagen in the mucosal region. Consistent with this, in a separate experiment in normal mice, we observed that ESM intake increased the type III/I collagen expression ratio in the colon, and we have added these data at the end of Results 3.
Comments 6: Results 4
Lines 234-237: The authors should discuss these as strong trends that, with a larger sample size, might become statistically significant.
Response to Comments 6 (Results 4):
As suggested, we have added a sentence indicating that these results represent a strong tendency, which might reach statistical significance with a larger sample size.
Comments 7: Discussion 1
Lines 319-321: The question of synergistic effects between LYZ and other ESM components is a good point for future research. I suggest that the authors should discuss this more thoroughly, emphasizing that while LYZ is a key component, ESM's full therapeutic effect is likely due to the combined action of its many proteins involved.
Response to Comments 7: Discussin1 (Lines 319–321):
As suggested, we have expanded the Discussion and added a paragraph emphasizing that although lysozyme (LYZ) is an important component, the full therapeutic effect of ESM is most likely attributable to the synergistic and combined actions of its many proteins.
“Although lysozyme (LYZ) is an important component, the overall therapeutic effect of ESM is likely attributable to the combined actions of its many proteins. ESM is composed of more than 700 proteins, and even those present at levels exceeding 1% are expected to have multiple functions. Moreover, ESM is fibrous, and as reported in our 2011 study {Ohto-Fujita, 2011 #64}, hydrolyzed ESM can self-assemble into a mesh-like structure, as demonstrated by AFM imaging. Interestingly, lysozyme is known to serve as a scaffold in eggshell formation. Therefore, fragments of ESM fibers or intestinal ECM may self-assemble around lysozyme and provide mechanical stimulation to cells, thereby activating cellular responses such as mitochondrial activity and HSP induction. Highly cross-linked by lysyl oxidase and possessing unique self-assembly properties (including potential phase separation), ESM represents a distinctive fibrous complex capable of exerting mechanostimulatory and mild stress effects. These aspects warrant further in-vestigation in future studies.”
Comments 8: Discussion 2
I suggest that the discussion should also include a paragraph on the limitations of the study, such as the small sample sizes in both, the animal and human trials, which impacts the statistical power of the findings.
Response to Comments 8 (Discussion 2):
As suggested, we have added a paragraph to the Discussion addressing the limitations of the study, specifically noting that the small sample sizes in both the animal experiments and the human trial may have impacted the statistical power of the findings.
Reviewer 3 Report
Comments and Suggestions for Authors
Dear Erudite Editors of the International Journal of Molecular Sciences and Esteemed Authors of the manuscript ijms-3804537, it is an honor to have been invited to review this critical submission to this most essential journal. Eggshell membrane and lysozyme 3 have shown potential in improving gut health during colitis. Eggshell membrane provides anti-inflammatory compounds that support gut barrier function and promote beneficial bacteria. Lysozyme 3 helps control harmful microbes, reducing inflammation and aiding microbiota balance. Together, they may help restore gut homeostasis and relieve colitis symptoms naturally. This submission is publishable. However, it is essential to address a few concerns with the Esteemed Authors before the journal decides whether to publish.
- Since this manuscript presents the results of an experimental trial, I recommend structuring the abstract using standard headings such as Background, Methods, Results, and Conclusions. A structured abstract would enhance clarity and allow readers to quickly grasp the key elements of the study design and findings.
- The manuscript would benefit from the inclusion of scientific illustrations to better communicate the study's findings and mechanisms. I recommend adding figures that visually summarize the proposed mechanisms of action: one schematic diagram showing how the eggshell membrane contributes to gut barrier integrity, reduces inflammation, and promotes beneficial microbiota, and another illustrating how lysozyme 3 helps suppress harmful microbes and supports microbiota balance. A combined figure highlighting the synergistic effect of both agents on restoring gut homeostasis during colitis would also enhance reader understanding and add visual impact to the manuscript. Tools such as BioRender, Mind the Graph, or Smart Servier Medical Art can be used to create high-quality, scientifically accurate illustrations tailored to these biological processes
- The authors are encouraged to rationalize better future research directions based on their current findings, clearly outlining how their results inform the next steps in experimental design or therapeutic development. Additionally, a more in-depth discussion of the potential clinical implications would strengthen the manuscript by helping readers understand how these findings could translate into real-world applications for managing colitis or promoting gut health
- I recommend the inclusion of a graphical abstract to visually summarize the main findings and mechanisms proposed in the study. A well-designed graphical abstract can enhance the manuscript’s accessibility, improve reader engagement, and facilitate understanding of the key concepts, especially for multidisciplinary audiences.
- Given that the study is a randomized, double-anonymized, placebo-controlled trial, I suggest including a dedicated table that addresses the potential sources of bias using a structured approach based on the Cochrane Risk of Bias framework. This table would significantly enhance the transparency of the methodology and allow readers to assess the study’s limitations and risk of bias more easily. I recommend structuring the table using the following headings: Question Focus, Appropriate Randomization, Allocation Blinding, Double-Blind, Losses (<20%), Prognostic or Demographic Characteristics, Outcomes, Intention-to-Treat Analysis, Sample Calculation, and Adequate Follow-Up. This addition would not only improve methodological rigor but also align the manuscript with best practices for reporting clinical trials.
- To improve the clarity and transparency of the study design, I recommend including a dedicated figure that outlines the screening, randomization, intervention, and analysis protocol. A well-structured flow diagram—such as a CONSORT-style flowchart—would help readers quickly grasp the progression of participants through each stage of the trial, from initial screening and eligibility assessment to group allocation, intervention delivery, follow-up, and data analysis. This visual summary would enhance the manuscript’s methodological transparency and support reproducibility.
- To ensure comprehensive and transparent reporting, I recommend that the authors incorporate all relevant elements of the CONSORT guidelines into the manuscript. This would significantly strengthen the methodological rigor and clarity of the trial presentation. If including the complete set of CONSORT elements is not feasible, I suggest providing a detailed table of prognostic and demographic characteristics for each study group. This would allow readers to assess baseline comparability and better understand potential confounding factors or sources of bias.
- To enhance the overall quality of the manuscript, I recommend improving the resolution and clarity of the figures, where the parameters are difficult to read and interpret due to low pixel quality. High-resolution images with clearly labeled axes, legible text, and well-defined graphical elements are essential to ensure that readers and reviewers easily understand key findings. Replacing low-quality images with publication-ready versions will significantly improve the visual impact and accessibility of the data.
- Although a conclusion section may not be strictly required, I highly recommend including one in this manuscript. A clear and concise conclusion would help to summarize the key findings effectively, emphasize their significance, and reinforce the overall message of the study for the readers.
- I also highly recommend that the methodology section be structured into numbered subsections. This will help readers complete their assessment of he manuscript with a standardized format.
- Authors must elucidate the limitations of their study in a separate subsection within the discussion. Also, the strengths should be further discussed, including the authors' collection of microbiota samples before the intervention starts.
- This manuscript has a similarity index of 26% after initial screening. I suggest that the authors lower this similarity index before reassessment.
Thank you for your patience and cooperation. I'm looking forward to receiving a revised version of this manuscript soon.
With best regards,
The Reviewer.
Author Response
We sincerely appreciate the Editor and Reviewers for their insightful comments and valuable suggestions, which have greatly helped us to improve our manuscript. We have carefully addressed each comment, revised the manuscript accordingly, and highlighted the changes in the revised version. We believe that these revisions have enhanced both the scientific rigor and readability of the paper. Detailed point-by-point responses are provided below.
Comments 1: Since this manuscript presents the results of an experimental trial, I recommend structuring the abstract using standard headings such as Background, Methods, Results, and Conclusions. A structured abstract would enhance clarity and allow readers to quickly grasp the key elements of the study design and findings.
Response to Comments 1 (Abstract):
As suggested, we have revised the abstract into a structured format with the standard headings Background, Methods, Results, and Conclusions to improve clarity and allow readers to readily grasp the key elements of the study.
Comments 2: The manuscript would benefit from the inclusion of scientific illustrations to better communicate the study's findings and mechanisms. I recommend adding figures that visually summarize the proposed mechanisms of action: one schematic diagram showing how the eggshell membrane contributes to gut barrier integrity, reduces inflammation, and promotes beneficial microbiota, and another illustrating how lysozyme 3 helps suppress harmful microbes and supports microbiota balance. A combined figure highlighting the synergistic effect of both agents on restoring gut homeostasis during colitis would also enhance reader understanding and add visual impact to the manuscript. Tools such as BioRender, Mind the Graph, or Smart Servier Medical Art can be used to create high-quality, scientifically accurate illustrations tailored to these biological processes
Response to Comments 2:
We appreciate the reviewer’s suggestion regarding the inclusion of scientific illustrations. At present, however, our data on the barrier effects of ESM in the intestine are limited, showing only tendencies, and the discussion of its major components remains largely hypothetical. Therefore, we believe it would be more appropriate to develop such schematic figures once additional experimental data are available. We will address this important point in future work.
Comments 3: The authors are encouraged to rationalize better future research directions based on their current findings, clearly outlining how their results inform the next steps in experimental design or therapeutic development. Additionally, a more in-depth discussion of the potential clinical implications would strengthen the manuscript by helping readers understand how these findings could translate into real-world applications for managing colitis or promoting gut health.
Response to Comments 3:
As suggested, we have expanded the Discussion to better outline future research directions and potential clinical implications. In a new section entitled Insights into translational research, we emphasize that proteomic analysis identified lysozyme (LYZ, ~50%) and lysyl oxidase as major protein components of ESM, and oral administration of tritium-labeled ESM demonstrated digestion, absorption, and systemic distribution, providing a rationale for its multi-organ effects. In the context of IBD, we addressed the knowledge gap regarding the in vivo effects of ESM in humans by analyzing fecal microbiota, a noninvasive biomarker. Our findings demonstrated improved microbial diversity, and in a double-blind, placebo-controlled trial, significant increases in beneficial bacteria were observed at weeks 4 and 8. Taken together, these results provide a basis for future studies to evaluate ESM supplementation as a potential preventive and therapeutic strategy for colitis and gut health.
Comments 4: I recommend the inclusion of a graphical abstract to visually summarize the main findings and mechanisms proposed in the study. A well-designed graphical abstract can enhance the manuscript’s accessibility, improve reader engagement, and facilitate understanding of the key concepts, especially for multidisciplinary audiences.
Response to comments 4
Thank you for your suggestion. We have prepared and included a graphical abstract to visually summarize the main methods and findings of the study.
Response to comments 5
We sincerely appreciate the reviewer’s thoughtful suggestion. In response, we created a structured table addressing potential sources of bias based on the Cochrane Risk of Bias framework. This table includes key items such as randomization, blinding, attrition, and outcome assessment. The table has been included as Table 1. 3. 4 in the Supplementary File. To enhance transparency, we have also added a sentence in the main text to guide readers to this supplementary material.
Furthermore, to improve clarity and transparency in the main text, we added a paragraph at the end of the Results, Section 2.4. Randomized Controlled Trial Assessing Gut Microbiota Response to Eggshell Membrane, which summarizes our responses to Comments 5, 6, and 7. The added paragraph is as follows:
“To enhance methodological transparency and ensure reproducibility, we incorporated relevant elements of the CONSORT guidelines where applicable. A structured risk of bias assessment based on the Cochrane framework is provided in Supplementary Table 1, covering key domains such as randomization, blinding, attrition, outcome measures, and follow-up. A CONSORT-style flow diagram summarizing the participant progression throughout the study is presented in Supplementary Figure 1. Additionally, a completed CONSORT checklist in Supplementary Table 2 and a table comparing prognostic and demographic characteristics between groups are included in Supplementary Table 3. The trial was prospectively registered in the UMIN Clinical Trials Registry (UMIN-CTR, No. 000057589).”
Comments 6: To improve the clarity and transparency of the study design, I recommend including a dedicated figure that outlines the screening, randomization, intervention, and analysis protocol. A well-structured flow diagram—such as a CONSORT-style flowchart—would help readers quickly grasp the progression of participants through each stage of the trial, from initial screening and eligibility assessment to group allocation, intervention delivery, follow-up, and data analysis. This visual summary would enhance the manuscript’s methodological transparency and support reproducibility.
Response to comment 6
Thank you for your valuable suggestion. In response to Comment 6, we created a CONSORT-style flowchart that outlines the screening, randomization, intervention, and analysis process. This flow diagram has been included as Supplementary Figure 1. We believe this visual summary improves the clarity and transparency of the study design and facilitates readers’ understanding of the participant flow throughout the trial.
Furthermore, to improve clarity and transparency in the main text, we added the paragraph at the end of the Results, Section 2.4. Randomized Controlled Trial Assessing Gut Microbiota Response to Eggshell Membrane.
Comments 7: To ensure comprehensive and transparent reporting, I recommend that the authors incorporate all relevant elements of the CONSORT guidelines into the manuscript. This would significantly strengthen the methodological rigor and clarity of the trial presentation. If including the complete set of CONSORT elements is not feasible, I suggest providing a detailed table of prognostic and demographic characteristics for each study group. This would allow readers to assess baseline comparability and better understand potential confounding factors or sources of bias.
Response to Comments 7
Thank you for your constructive feedback. We agree that adherence to CONSORT guidelines enhances the transparency and methodological rigor of clinical trial reporting. Given that the primary focus of this study is based on a life-science approach involving both human and animal experiments, we have incorporated relevant CONSORT elements into the manuscript where appropriate. In addition, we have provided a completed CONSORT checklist as Supplementary Table 2 and included a table summarizing prognostic and demographic characteristics for each study group to facilitate evaluation of baseline comparability (see Supplementary Table 3).
To address items included in the CONSORT checklist, we added supplementary descriptions in the Methods section, specifically in the middle and at the end of the Human Study subsection.
The added paragraphs are as follows:
Human Study
This study was conducted concurrently with our previous report, which evaluated functional outcomes including skin elasticity, respiratory function, and physical performance [39]. Twenty healthy, non-smoking male and female volunteers (age range: 22–68 years; mean ± SD: 37.1 ± 12.6 years) who agreed to participate in the study were enrolled. The study was conducted at Tokyo University of Agriculture and Technology over 8 weeks from September to November 2015, with participant recruitment carried out during a two-week period in August 2015.
…… (omitted) ……
Participants were instructed to refrain from taking other dietary supplements or undergoing body aesthetic treatments during the study period. The supplement used in this study was considered safe. In the event of adverse effects related to the supplement or study procedures, participants were advised to seek prompt medical attention, and appropriate treatment and compensation were guaranteed if necessary.
Comments 8: To enhance the overall quality of the manuscript, I recommend improving the resolution and clarity of the figures, where the parameters are difficult to read and interpret due to low pixel quality. High-resolution images with clearly labeled axes, legible text, and well-defined graphical elements are essential to ensure that readers and reviewers easily understand key findings. Replacing low-quality images with publication-ready versions will significantly improve the visual impact and accessibility of the data.
Response to Reviewer 8:
We appreciate the reviewer’s comment. To address this, we will upload high-resolution versions of all figures as separate files, ensuring clearly labeled axes, legible text, and publication-ready quality to improve clarity and visual impact.
Comments 9: Although a conclusion section may not be strictly required, I highly recommend including one in this manuscript. A clear and concise conclusion would help to summarize the key findings effectively, emphasize their significance, and reinforce the overall message of the study for the readers.
Response to Comments 9:
As recommended, we have added a Conclusion section to summarize the key findings and emphasize their significance. The new section reads as follows:
Conclusion
Proteomic analysis of the protein components of ESM, which is known to promote wound healing, rejuvenate the ECM, and attenuate fibrosis in the liver and lung, revealed that approximately 50% consisted of lysozyme (LYZ), followed by lysyl oxidase. Oral administration of tritium-labeled ESM demonstrated its digestion, absorption, and systemic distribution, providing a rationale for the multi-organ effects of ESM. In the context of IBD, we addressed the knowledge gap regarding the in vivo effects of ESM in humans by focusing on fecal microbiota, which can be assessed noninvasively. Our results showed that ESM intake improved microbial diversity, and in a double-blind, placebo-controlled trial in healthy adults, significant increases in beneficial bacteria were observed at weeks 4 and 8. Taken together, these findings suggest that ESM supplementation holds promise as a novel preventive approach for intestinal diseases with unmet medical needs.
Comments 10: I also highly recommend that the methodology section be structured into numbered subsections. This will help readers complete their assessment of he manuscript with a standardized format.
Response to Comments 10:
As recommended, we have revised the Methodology section and structured it into numbered subsections to provide a clearer and more standardized format.
Comments 11: Authors must elucidate the limitations of their study in a separate subsection within the discussion. Also, the strengths should be further discussed, including the authors' collection of microbiota samples before the intervention starts.
Response to Comments 11:
As requested, we have added a separate subsection in the Discussion to address the limitations and strengths of our study. The primary limitation is the small sample size in both the animal experiments and the human trial, which may have restricted the statistical power. On the other hand, the strengths include: (i) proteomic characterization of the ESM protein components prior to intervention, thereby clarifying the major functional constituents for readers; (ii) the use of a unique labeling method that enabled tritium labeling of insoluble composite proteins, allowing oral administration to mice in a manner comparable to human supplementation; and (iii) demonstration that, beyond the transient direct effects on gut microbiota as observed in fecal absorption peaks, absorbed components were distributed via the bloodstream to the colon and other organs. These findings complement our previous reports on the effects of ESM supplementation in the skin and lung, while bridging the knowledge gap on its intestinal effects in the context of IBD by showing consistent microbiota changes in both mice and humans.
Comments 12: This manuscript has a similarity index of 26% after initial screening. I suggest that the authors lower this similarity index before reassessment.
Thank you for your comment. Although we do not have access to a similarity checking tool, we carefully revised the manuscript and believe that the originality has been improved in this revision.
Reviewer 4 Report
Comments and Suggestions for Authors
Thank you for submitting your article for consideration at IJMS.
The idea is sound but here are my comments
The introduction section need to be enriched to cover this gap of knowledge
A graphical abstract is highly recommended to explain the main point of your study
Result section, this is the best part, is well designed and represented
Where's the study limitation section
In figure 2, where is the kidney data, being a major execretory organ in addition to the liver
What is the reason behind using this dose of powdered ESM, have you tried a higher or lower dose?
References need to decreased a little bit
English needs to be revised as some typos were detected
Comments on the Quality of English Language
English needs to be revised as some typos were detected
Author Response
We sincerely appreciate the Editor and Reviewers for their insightful comments and valuable suggestions, which have greatly helped us to improve our manuscript. We have carefully addressed each comment, revised the manuscript accordingly, and highlighted the changes in the revised version. We believe that these revisions have enhanced both the scientific rigor and readability of the paper. Detailed point-by-point responses are provided below.
Comments 1: The introduction section need to be enriched to cover this gap of knowledge
Response to Comment 1:
We have revised the Introduction and added the point regarding the knowledge gap on the in vivo effects of ESM in humans in the context of IBD.
Comments 2: A graphical abstract is highly recommended to explain the main point of your study
Response to Comments 2:
As recommended, we have prepared a graphical abstract that highlights the main point of our study, illustrating how it bridges the gap between experimental findings and clinical relevance.
Comments 3: Result section, this is the best part, is well designed and represented
Response to Comments 3:We sincerely thank the reviewer for the positive comment on the Results section.
Comments 4: Where's the study limitation section
Response to Comments 4:
We have added a subsection on the study limitations at the end of the Discussion.
Comments 5: In figure 2, where is the kidney data, being a major execretory organ in addition to the liver
Response to Comments 5:
As suggested, we also examined the kidney, a major excretory organ in addition to the liver. We confirmed an early absorption peak followed by gradual excretion. Accordingly, we added the following description to the Results section: “Interestingly, the kidney showed an early peak at 2 h, preceding the serum peak at 5 h, and was likely reabsorbed and gradually excreted thereafter. This observation is intriguing because inhibition or blockade of megalin-mediated endocytosis of nephrotoxic substances has been reported as a potential therapeutic strategy against drug-induced nephrotoxicity and metabolic kidney diseases.”
Comments 6: What is the reason behind using this dose of powdered ESM, have you tried a higher or lower dose?
Response to Comments 6:
ESM is consumed as a dietary supplement by healthy individuals, and therefore we first conducted a human study. For the mouse experiments, the dose was determined by converting the amount typically ingested by humans to an equivalent dose based on body weight. Other doses were not tested in this study. We also note that in our previously published study using a bleomycin-induced pulmonary fibrosis mouse model, the same dose of ESM was administered and shown to ameliorate disease progression.
Comments 7: References need to decreased a little bit
Response to Comments 7:
As also suggested by another reviewer, we have substantially reduced the number of references in the Introduction. Overall, the total number of references has been decreased from 77 to 56.
Comments 8: English needs to be revised as some typos were detected
Response to Comments 8:
We thank the reviewer for pointing this out. We have carefully revised the manuscript to correct the typos and improve the overall English.
Round 2
Reviewer 1 Report
Comments and Suggestions for Authors
The authors have revised the manuscript as requested
Author Response
We sincerely appreciate the reviewer’s acknowledgment. We are grateful for the constructive comments, which have helped us to substantially improve the manuscript.
Reviewer 3 Report
Comments and Suggestions for Authors
Dear Authors, thank you for your revisions. I believe that the manuscript is in good format. However, there are a few other suggestions that I would like to address with you before your manuscript gets a final decision by the Editor.
- Your abstract is too lengthy. While I acknowledge that you structured your abstract to ensure rigor and transparency in presenting your findings, I suggest that decreasing its length would significantly enhance the article’s in-depth importance in the field. Try letting the results stay and diminishing the introduction, methods, and conclusions content to only the utmost necessary information.
- Lines 348-447 lack sufficient attention to the format. Ensure that all paragraphs are formatted in a structured manner throughout the entire manuscript.
- Lines 436-447 aren’t formatted into a section. Please ensure all sections are adequately headed within the main document of your manuscript.
- The content of Figures 6-8 isn’t easy to read. Please ensure high-quality figures are presented within your manuscript's main document.
- Include in your main document a brief table that describes the ARRIVE checklist for your in vivo experiments. This will allow readers to address possible bias in your experiments promptly. Do the same as the COCHRANE handbook for your randomized, double-blind experiments. For the tables, please use standard headings following each methodology.
- Rationalize better the inclusion and exclusion criteria of your study in your methodology section.
Thank you for your attention to these details.
Author Response
We sincerely thank Reviewer 3 (Round 2) for the positive evaluation of our revised manuscript and for the additional helpful suggestions. We have carefully considered each of the points raised and revised the manuscript accordingly, as detailed below.
[Reviewer comment 1]
Your abstract is too lengthy. While I acknowledge that you structured your abstract to ensure rigor and transparency in presenting your findings, I suggest that decreasing its length would significantly enhance the article’s in-depth importance in the field. Try letting the results stay and diminishing the introduction, methods, and conclusions content to only the utmost necessary information.
Response to comment 1
We thank the reviewer for this valuable suggestion. In accordance with the recommendation, we have revised the abstract by shortening the introduction, methods, and conclusion sections, while retaining the essential results to ensure clarity and conciseness.
■Line 30
Background Eggshell membrane (ESM), composed of approximately 90% protein, has been shown to provide a type III collagen–rich extracellular matrix (ECM) environment in the skin and to reduce lung fibrosis when applied to fibroblasts and mice, partly through TGF-β–sequestering and enhanced decorin secretion. Our previous studies in healthy adults also demonstrated that two months of daily ESM intake improved respiratory function, zigzag walking speed, and skin elasticity. The present study aims to address the knowledge gap regarding the in vivo effects of ESM in the context of inflammatory bowel disease (IBD).
■Line 38
Methods Proteomic analysis was performed on powdered ESM used as a dietary supplement. To investigate its pharmacokinetics in mice, tritium (³H)-labeled ESM was prepared using the ⁶Li(n,α)³H nuclear reaction. For radioactivity measurement, tissues were solubilized with SOLVABLE™, mixed with a scintillation cocktail, and analyzed using a liquid scintillation counter. The therapeutic potential of ESM was further examined in a 2.0% dextran sulfate sodium (DSS)-induced murine model of inflammatory bowel disease (IBD). In addition, fecal samples from both mice and healthy human subjects were analyzed using a modified terminal restriction fragment length polymorphism (T-RFLP) method.
■Line 64
Conclusions ESM supplementation demonstrated favorable pharmacokinetic distribution and exerted protective effects against DSS-induced colitis in mice, while also promoting beneficial modulation of gut microbiota in both murine and human studies. These findings suggest that ESM, through its major protein components such as lysozyme C LYZ and lysyl oxidase, may serve as a promising dietary intervention for maintaining intestinal health and mitigating inflammation in the context of IBD.
[Reviewer comment 2]
Lines 348-447 lack sufficient attention to the format. Ensure that all paragraphs are formatted in a structured manner throughout the entire manuscript.
Response to comment 2
■In accordance with the reviewer’s advice, we have standardized the formatting in the Discussion section, including Lines 348–447, lines Introduction section (126-138), Discussion section (line 320-416), Conclusion section, Materials and Methods section, by adjusting kerning and removing excessive paragraph spacing.
■Lines 260–261 (legend for Figure 5): The bacterial genus names have been italicized (i.e., Bacteroidales and Lactobacillus).
[Reviewer comment 3]
Lines 436-447 aren’t formatted into a section. Please ensure all sections are adequately headed within the main document of your manuscript.
Response to comment 3
In accordance with the reviewer’s advice, we have added numbering to the Conclusion section, which is now designated as Section 4 (line 437). Consequently, the Methods section (line 449- 661) has been renumbered as Section 5 to maintain consistency throughout the manuscript. According to the submission guidelines, section numbers should not be assigned to the Supplementary files list; therefore, we have removed them (line 562).
[Reviewer comment 4]
The content of Figures 6-8 isn’t easy to read. Please ensure high-quality figures are presented within your manuscript's main document.
Response to comment 4
We have prepared publication-grade, high-resolution figures (Figure 5~9) and replaced them into the revised manuscript (R2 version).
[Reviewer comment 5]
Include in your main document a brief table that describes the ARRIVE checklist for your in vivo experiments. This will allow readers to address possible bias in your experiments promptly. Do the same as the COCHRANE handbook for your randomized, double-blind experiments. For the tables, please use standard headings following each methodology.
Response to comment 5
We are grateful for this insightful suggestion. In accordance with the reviewer’s advice, we have undertaken the following revisions:
■Line 663: For clarity, a concise table summarizing the ARRIVE checklist has been included as Supplementary Table 1.
■Section 2.2 line 147
Following sentence was added.
For clarity, a concise table summarizing the ARRIVE checklist has been included as Supplementary Table 1.
■Section 2.3 line 187
Following sentence was added.
For clarity, a concise table summarizing the ARRIVE checklist has been included as Supplementary Table 1.
■Section 2.4 Line298 We have revised our risk of bias assessment to follow the Cochrane Handbook (RoB2) format, using the standard domains and judgments. The table, including detailed supporting information, has been provided as Supplementary Table 2 (R2). In the main text, we added a sentence to guide readers to this supplementary material. We believe this revision improves the methodological rigor and transparency of our reporting.
■Section 2.4 Since Supplementary Table 1 has been newly added, the previous Supplementary Tables 2-4 have been renumbered as Supplementary Tables 3-5. In R1, we realized that the citations for Supplementary Tables 4a and 4b were missing in the main text; we have now inserted them (Supplementary Tables 5a and 5b in R2) accordingly.
Line 294-304 in manuscript (R2)
A risk of bias assessment was conducted in accordance with the Cochrane Handbook (RoB2). The results are summarized in Supplementary Table 2, covering domains such as randomization, deviations from intended interventions, missing data, outcome measurement, and reporting. A CONSORT-style flow diagram summarizing the participant progression throughout the study is presented in Supplementary Figure 1. Additionally, a completed CONSORT checklist and a table comparing prognostic and demographic characteristics between groups are included in Supplementary Table 3 and Supplementary Table 4, respectively. Post-hoc power analyses and effect sizes were calculated using G*Power 3.1 to evaluate the statistical validity of the findings (see Supplementary Table 5a,b). The trial was prospectively registered in the UMIN Clinical Trials Registry (UMIN-CTR, No. 000057589).
[Reviewer comment 6]
Rationalize better the inclusion and exclusion criteria of your study in your methodology section.
Response to comment 6
■Method Section 5.7 Human Study
line 605
Twenty healthy, non-smoking male and female volunteers (age range: 22–68 years; mean ± SD: 37.1 ± 12.6 years) who agreed to participate in the study were enrolled.
Revised
Twenty healthy, non-smoking male and female volunteers (age range: 22–68 years; mean ± SD: 37.1 ± 12.6 years) who agreed to participate in the study were enrolled. Inclusion criteria required that participants had no notable current medical conditions at the time of enrollment. Exclusion criteria included a history of food allergies, cardiovascular disease, hypertension, thyroid disorders, medical advice prohibiting exercise, and regular use of dietary supplements.
[Author comment to Reviewer 3]
In addition to the specific points noted in your comments, we have undertaken further revisions to enhance the overall quality of the manuscript, including unifying the use of abbreviations and supplementing missing information. These additional revisions are listed below.
Line 6: Added * to Shimizu to indicate correspondence.
Line 70 (Keywords section): Added abbreviation for lysozyme (LYZ).
Lines 101, 131, 192, 353, 358, 359, 410, 445: Replaced “lysozyme” with “LYZ.”
Line 126: Replaced “eggshell membrane” with “ESM.”
Line 169-170 (Figure 2): Changed layout from 2×4 +1 horizontal panels to 3×3 panels for improved appearance.
Lines 179-205; 216- 226; Line 275: Increased font size from 9 to 10 to match main text.
Lines 268-270, 292, 368-370, 649: Italicized bacterial group names.
Line 312: Added abbreviation “eggshell membrane (ESM)” in stand-alone caption.
Line 329: Added “kidney” (previously missing).
Lines 332, 347, 470, 487, 489: Inserted half-space for consistency.
Line 340: Used abbreviation “DAI” instead of full term “disease activity index.”
Line 426: Included names of primary and secondary antibodies used.
Line 500: Italicized in vivo.
Line 527: Used abbreviations for ESM and LYZ, and added each dose.
Line 585: Standardized wording to “fine ESM powder” instead of “micro.”
Line 620: Added dose information.
Line 703-704 (Abbreviations): Added “DAI.”
Revised graphical abstract: Added dose information to strengthen our experimental design, emphasizing the concept of “filling the gap between mice and humans.”